# Changes in Affective Behavior and Oxidative Stress after Binge Alcohol in Male and Female Rats

**DOI:** 10.3390/brainsci11091250

**Published:** 2021-09-21

**Authors:** Ibanelo Cortez, Patricia S. Brocardo, J. Leigh Leasure

**Affiliations:** 1Department of Psychology, University of Houston, Houston, TX 77204, USA; cortezdaneloish@gmail.com; 2Department of Morphological Sciences and Graduate Neuroscience Program, Center of Biological Sciences, Federal University of Santa Catarina, Florianopolis 88040-900, SC, Brazil; 3Department of Biology & Biochemistry, University of Houston, Houston, TX 77204, USA

**Keywords:** sex differences, glutathione, lipid peroxidation, elevated plus maze, forced swim test, open field

## Abstract

Binge alcohol consumption and alcohol use disorders (AUD) are prevalent, and there is comorbidity with depression and anxiety. Potential underlying mechanisms include neurophysiological, genetic, and metabolic changes resulting from alcohol exposure. Mood and anxiety disorders are more common among women, but whether females are more susceptible to binge-induced oxidative stress and co-occurring anxiety and depression-like behaviors remains unknown. Here, we used a repeated, weekly binge alcohol paradigm in male and female rats to investigate sex differences in despair and anxiety-like behaviors and brain oxidative stress parameters. A single binge alcohol exposure significantly elevated glutathione (GSH) levels in prefrontal cortex (PFC) of both male and female animals. This was accompanied by increased lipid peroxidation in PFC of both sexes. Repeated (once weekly) binge exposure induced changes in anxiety- and depression-like behaviors in both males and females and increased GSH level in the PFC without detectable oxidative damage. Our findings suggest that repeated binge alcohol exposure influences affect regardless of sex and in the absence of membrane damage.

## 1. Introduction

Binge drinking is the most common pattern of alcohol intake in the United States [1]. By definition, it produces a blood alcohol content of >0.08 g/dL, which equates to approximately 5+ drinks for men and 4+ for women [2]. Binge drinking increases the risk of developing alcohol use disorder (AUD) and differs from chronic alcohol use in intensity and frequency of consumption. Binge drinkers consume alcohol infrequently yet heavily in a single episode, while chronic alcohol drinkers consume moderate to heavy amounts daily. Although alcohol dependence is not typically observed in binge drinkers [3], this pattern of consumption can be detrimental to cognition and mood [4]. Binge alcohol has also been shown to be neurotoxic in both humans and animal models (see [5] for a recent review) and is linked to negative effects on mental health and may trigger or exacerbate mood disorders [6]. Affective disorders, like anxiety and depression, are more common in women, and women are more likely to binge drink to regulate mood disorders [7,8]. Furthermore, young men who report suffering from anxiety and depression are also likely to binge drink [6]. Thus, although binge drinking may not induce pathologies associated with withdrawal, studies detect deficits in cognitive and emotional domains that can still interfere with daily activities.

Recent evidence indicates that binge drinking is increasing among women [9]. In light of this finding, research has begun to elucidate differences between male and female binge drinkers. Females that report binge drinking in the last month are more susceptible compared to non-binge drinkers to cognitive/motor deficits [10,11,12], changes in neural activation [13,14,15], and white matter loss [16]. Moreover, some studies have reported that females are more susceptible to present acute and long-term alterations of mood and memory [17] and morphological changes [11] after binge drinking compared to males. Beyond that, oxidative balance also seems to be different between males and females. For example, Jung and Metzger [18] found an oxidative stress sex difference after alcohol exposure.

During the process of alcohol metabolism, toxic and highly reactive products, such as acetaldehyde and reactive oxygen species (ROS), are generated [19]. In the brain, alcohol metabolism generates acetaldehyde and an excess of ROS, resulting in oxidative damage, which is one of the main mechanisms of tissue injury [20,21]. In fact, the brain is particularly vulnerable to the production of ROS since it metabolizes 20% of the total oxygen in the body and has a limited amount of antioxidant capacity [22,23,24]. Endogenous antioxidant defenses include multiple enzymes and their co-factors, which can either inhibit the formation of ROS or promote the removal or scavenging of free radicals and their precursors. The major antioxidant enzymes directly involved in the neutralization of ROS are superoxide dismutase (SOD), catalase (CAT), glutathione peroxidase (GPx), glutathione reductase (GR), and glutathione-S-transferases (GSTs). Glutathione (GSH) is the most important non-enzymatic endogenous antioxidant and can be regenerated by GR with the consumption of nicotinamide adenosine dinucleotide phosphate (NADPH) [25]. Importantly, GSH is also a cofactor for the redox reactions catalyzed by GSTs, which play a role in detoxifying a variety of electrophilic xenobiotics, producing less toxic compounds [26].

Developmental exposure to alcohol induces membrane damage and decreased GSH levels in adult animals [27]. Moreover, evidence for oxidative stress as a mediator of proinflammation that leads to neurodegeneration during intoxication is well documented [28]. In a four-day binge alcohol model, oxidative stress stimulated proinflammatory gene expression concomitant with gliosis, neurodegeneration, and memory impairment [29]. Several studies have indicated that oxidative stress also plays a major role in the etiology of depression and anxiety [30] and that these mood disorders are accompanied by an increase in markers of oxidative stress [31,32,33] and a concomitant decrease in the endogenous antioxidant defenses (for review, see [28,34]). Using different inbred strains of mice and lentivirus-mediated gene transfer to overexpress glyoxalase 1 and glutathione reductase 1 genes in the mouse brain, Hovatta and colleagues found that these genes have a causal role in the genesis of anxiety. Both of these genes are involved in oxidative stress metabolism, linking this pathway with anxiety-related behavior [35]. Relatedly, sub-chronic induction of these oxidative genes reduced BDNF expression and increased pro-inflammatory genes, such as IL-6, TNF-α, and angiotensin [36].

Sex hormones may also play a role in generating oxidative stress, which is speculated to be the reason for sex disparities in certain neurodegenerative diseases [37]. Moreover, oxidative stress influences behavior and can be a trigger for neuroimmune responses that can promote neurodegenerative mechanisms. Considering that binge alcohol exposure leads to oxidative stress and that increased oxidative stress may be a predisposing factor for anxiety and depression [38,39], in the present study, we evaluated the occurrence of anxiety- and depressive-like behaviors and the levels of oxidative stress in a rat model of binge alcohol and whether these occurred in a sex-dependent manner. We predicted that female subjects would exhibit higher anxiety/depressive-like behaviors and increased oxidative stress and GSH levels and decrease in glutathione-related antioxidant enzymes activities.

## 2. Materials and Methods

### 2.1. Animals

All experimental procedures were conducted in accordance with the Guide for the Care and Use of Laboratory Animals of the National Institutes of Health. The University of Houston Institutional Animal Care and Use Committee approved the procedures (protocol number 16-013). Male and female Long–Evans rats were purchased at approximately 2 months of age (Envigo, Indianapolis, IN, USA), with males weighing 250–300 g and females weighing 175–200 g. Upon arrival, animals were allowed to acclimate to reverse light/dark_(12h2h)_ room vivarium conditions for one week. The following week, rats were gently handled daily for 5 days and trained to accept a gavage needle before treatments commenced.

### 2.2. Alcohol Administration Paradigm

Young adult rats were used in order to model human behavior considering that binge drinking is most common among young adults aged 18–34 years [40]. The animals were separated into three groups: alcohol binge, binge control, and naïve. Within these groups, a total of 23 (11 males) rats were treated with 5 g/kg of ethanol diluted in Ensure^®^-water solution (40% *w*/*v*) once a week for 4 weeks, as we have previously shown that 3 or more weekly binge exposures is associated with significantly fewer neurons in the hippocampal dentate gyrus [41]. Eighteen binge control rats (9 males) were treated with an isocaloric Ensure^®^-glucose solution, and 18 naïve rats (9 males) were handled daily and weighed once a week. All subjects were behaviorally tested 48 h after the third binge and sacrificed 1 h after the fourth binge treatment for antioxidant and oxidative damage markers (Figure 1). For measurement of blood ethanol concentration (BEC), whole blood was collected 60–90 min after each binge treatment from the saphenous vein. Whole blood was spun down at 600× *g* for 15 min, and plasma was collected and stored at −4 °C until all samples were processed using a GM7 Analyzer (Analox, Toronto, Canada).

In addition, a separate group of rats (8 males and 8 females, 6 binged and 2 naïve for each sex) was used for an initial pilot study in which we verified that a 5 g/kg binge administration of alcohol impacted GSH and membrane damage. For this initial statistical analysis, the naïve group included also naïve controls used in the main experiment. Binged animals in this pilot experiment received only one binge dose and were not behaviorally tested.

### 2.3. Behavioral Testing

Rats underwent the following behavioral tests: (1) open-field test (OF), to assess their locomotor activity; (2) elevated-plus maze (EPM) test, to evaluate the occurrence of anxiety-like behaviors; and (3) the forced swim test (FST), in order to assess the occurrence of depressive-like behaviors. For open-field and EPM tests, subjects were brought into testing rooms and allowed to acclimate for 30 min prior to testing. To minimize stress carry- over effects, testing began with OF (least stressful) and ended with FST. The habituation and test phases of FST were carried out in a separate room from the other tests. All testing occurred during the animals’ dark cycle. The open-field and EPM testing were carried out under red lighting, while FST was illuminated with fluorescent lighting.

Open Field: Subjects were randomized by group and sex and tested for natural exploration in a OF test. Briefly, rats were placed in the center of the OF and to allowed to freely explore a 58 cm × 58 cm × 39 cm (W × L × H) open field box for 30 min. Distance, velocity, and time spent in center/perimeter arenas were recorded using the Noldus video tracking software, Ethovision XT^®^ (Noldus, Amsterdam, The Netherlands). Natural exploration behavior was assessed by recording distance traveled and time spent exploring the center of the OF box. Open-field behavior for each subject was analyzed and graphed using the first 5 min of recording.

Elevated Plus Maze: A rodent’s natural tendency to explore safe and risky spaces can be observed using the EPM, which is composed of open arms and closed arms with raised walls connected by a center arena raised 33 cm from the ground and commonly used to test for anxiety-like behaviors [42]. After animals completed open-field testing, they were placed in the EPM, and Noldus video tracking software-Ethovision XT was used to measure total distance, velocity, entries, and time spent in open, closed, and center zones. Video recording was accomplished under red lighting with a fixed aerial camera a few feet above the EPM apparatus. To track performance, small dots generated by Noldus software were overlayed on the rat’s head and back, between the shoulders. Anxiety-like behavior was assessed by comparing % time spent (duration in zone/300 s) and frequency to enter open/closed arms and center zones. Head entries were counted if the dot overlayed on the animal’s head fully crossed into the zone.

Forced Swim Test (FST): Binge drinking is associated with depressive symptomatology in individuals with comorbid AUD and depression [43,44,45], a condition that is difficult to treat [46]. To determine if repeated binge alcohol affects depressive-like behaviors, we tested animals in the FST. In rodent experiments, behavioral despair, which can be assessed as immobility in the FST, is interpreted as a depressive-like behavior, as antidepressant drugs consistently reduce immobility in this behavioral test [47,48,49]. Subjects were tested in the forced swim assay after other behavioral tests were completed. Two 43 cm cylinders filled with tap water (23 °C ± 1 °C) 10 cm from the brim were used. The test consisted of two phases: a 15-min habituation phase followed by a 5-min test phase 24 h later. The habituation phase is a 15-min task which exposes subjects to test conditions and forces animals to develop a swimming strategy to stay afloat. Depression-like behavior was assessed during the 5-min test phase, which occurred 24 h after the habituation phase. All subjects were video-recorded and manually scored during the testing phase. Time-spent immobile was scored at times when subjects were completely immobile (no limb movement). Vertical climbing was scored when animal’s forelimbs broke the surface of the water in the center and against the wall of the cylinder. Lastly, horizontal swimming was scored as treading water, diving, or when the animal crossed the cylinder.

### 2.4. Corticosterone Measurements

Sixty to ninety minutes after the final binge treatment, animals were overdosed with an anesthetic cocktail (ketamine/xylazine/acepromazine), and trunk blood was collected and held on ice for ~30 min in centrifuge tubes containing 10 ul of 0.5 M EDTA. Whole blood was spun down at 600× *g* for 15 min, and 100 μL was collected for corticosterone analysis, and the remaining plasma was stored at −4 °C for BEC measurement. The Enzo Life Sciences Elisa kit (Cat# ADI-901- 097) (Farmingdale, NY, USA) was used to analyze corticosterone levels in plasma samples. A 1:40 dilution (plasma/buffer) was used with in the recommended small serum/plasma protocol for this kit.

### 2.5. Tissue Preparation

Fresh frozen samples from both hemispheres of prefrontal cortex (PFC), hippocampus, and striatum used to determine the activity of endogenous antioxidant enzymes were dissociated with chilled pestles on ice in 600 μL of 20-mM Hepes, pH 7.2. Dissociated samples were homogenized by submerging tip of mechanical homogenizer set at 30,000 rpm into 1.5-mL centrifuge tubes for 3–5 s. Homogenizer was rinsed with 50% ethanol and water between samples. Samples were spun down at 10,000× *g* for 10 min at 4 °C. Next, 125 μL of supernatant was collected and stored at −80 °C for TBARS assays. Samples were spun down again at 15,000 *g* for 20 min at 4 °C, and remaining supernatant was collected for glutathione reductase, S-transferase, and peroxidase activity assays. Fresh brain tissue used to determine the total levels of glutathione were homogenized in cooled 0.5 M perchloric acid. The homogenates were centrifuged at 15,000 *g* for 2 min, and the supernatant was separated and neutralized in potassium phosphate buffer (0.1 M, pH 7.4). The total protein content of the samples was determined by the bicinchoninic acid (BCA) method using a BCA Protein Assay kit (Thermo Scientific, Rockford, IL, USA) and following the manufacturer’s instructions. Because each brain region was used for multiple assays, we were unable to include animals that did not have sufficient volumes for testing. For this reason, one female naive subject was excluded from all antioxidant measurements and TBARS analyses in the hippocampus. Additionally, a male and female naive subject were excluded from protein carbonyl analysis in every brain region. Lastly, in our acute binge study, a male binge and naive subject were excluded from TBARS analyses in the PFC.

### 2.6. Activity of Antioxidant Enzymes and Total Levels of Glutathione

Glutathione (GSH) level was assessed in discrete brain regions immediately following euthanasia. Using Akerboom and Sies (1981) [50] methods adapted from Tietze’s original protocol [51], total glutathione was measured by the reaction of the 5,50-dithiobis(2-nitrobenzoic acid) (DTNB)- GR recycling assay. The sulfhydryl group of GSH reacts with DTNB and produces a yellow-colored 5-thio-2-nitrobenzoic acid (TNB). GSTNB, the mixed disulfide formed between GSH and TNB, is reduced by glutathione reductase to recycle the GSH and produce more TNB, which was measured continuously at 412 nm. To determine the total GSH concentration in the samples, a linear regression to calculate the values obtained from the standard curve was used. Total GSH levels was corrected by weight of tissue.

Brain glutathione peroxidase (GPx) was assessed using a method designed by Wendel (1981) [52], using cumene hydroxide as a substrate. NADPH disappearance was monitored by a spectrophotometer at 340 nm. Brain glutathione reductase (GR) activity was measured using the method of Carlberg and Mannervik (1985) [53]. This assay is based on the reduction of oxidized glutathione (GSSG) by NADPH, which is catalyzed by GR. The concomitant oxidation of NADPH to NADP+ is accompanied by a decrease in absorbance at 340 nm that is directly proportional to the GR activity in the sample. Glutathione S-Transferase (GST) activity was assessed using a method designed by Habig and Jakoby (1981) [54] using 1-chloro-2,4-dinitrobenzene (CDNB) as substrate. CDNB is not specific for any particular GST isoenzyme, being able to react with a broad range of GST isoenzymes. The assay was conducted by monitoring the appearance of the conjugated complex of CDNB and GSH, which results in an increase in the absorbance at 340 nm.

### 2.7. Oxidative Damage Assays

Lipid peroxidation was assessed using the Cayman Chemical Thiobarbituric Acid Reactive Substances (TBARS) Assay Kit (cat# 10009055) (Ann Arbor, MI, USA) following the manufacturer’s instructions. The amount of malondialdehyde (MDA) produced was determined spectrophotometrically at 532 nm. Protein carbonylation was assessed using the Cayman Chemical Protein Carbonyl Colorimetric Assay Kit (Cat# 10005020) following the manufacturer’s instructions. This assay utilizes the DNPH reaction to measure the protein content in brain homogenates. The amount of protein-hydrozone produced was quantified spectrophotometrically at 362 nm at room temperature.

### 2.8. Statistical Analyses

BEC data for animals that underwent 4 binge exposures were analyzed using repeated measures ANOVA. Body weight data were analyzed separately for each sex (as a sex difference was expected) using one-way ANOVA. All other data were analyzed using two-way ANOVA for the factors sex and treatment (naïve, control, and binge). Biochemical data were first converted to % male naive values. When appropriate, Tukey’s post-hoc comparisons were performed. Throughout, a *p*-value < 0.05 was deemed significant. Statistical analyses were performed using GraphPad Prism 7 (Graphpad Software, San Diego, CA, USA). In the figures, data are expressed as mean ± standard error of the mean. The ROUT outlier test was used to identify and remove a single data point from the binge female group, which had a more than two-fold-higher corticosterone level than the next highest value.

## 3. Results

### 3.1. Single Binge

To verify that binge alcohol exposure impacts antioxidant levels and membrane damage, we performed an initial experiment in which we examined GSH levels and oxidative damage in untreated (naïve) animals and those that were exposed to a single 5 g/kg binge dose (Figure 2). In the PFC, we observed a significant main effect of treatment (F(1, 30) = 36.0, *p* < 0.05) but not sex (F(1, 30) = 0.40, *p* = 0.52) and no significant interaction (F(1, 30) = 0.0, *p* = 0.97) on GSH levels. In the hippocampus, there was a significant main effect of treatment (F(1, 30) = 8.27, *p* < 0.05) but not sex (F(1, 30) = 0.50, *p* = 0.48) and no significant interaction (F(1, 30) = 1.05, *p* = 0.31). In the striatum, there was no significant main effect of treatment (F(1, 30) = 2.3, *p* = 0.14) or sex (F(1, 30) = 2.92, *p* = 0.10) and no significant interaction (F(1, 30) = 1.91, *p* = 0.18) on GSH levels.

We also assessed lipid peroxidation (TBARS). In the PFC, there was a significant main effect of treatment (F(1, 28) = 26.21, *p* < 0.05) but not sex (F(1, 28) = 1.88, *p* = 0.18) and no significant interaction (F(1, 28) = 0.07, *p* = 0.78). In the hippocampus, there were no significant main effects of treatment (F(1, 28) = 0.01, *p* = 0.90) or sex (F(1, 28) = 0.20, *p* = 0.65) and no significant interaction (F(1, 28) = 0.92, *p* = 0.34). Similarly, in the striatum, there were no significant main effects of treatment (F(1, 30) = 0.01, *p* = 0.93) or sex (F(1, 30) = 0.0, *p* = 0.94) and no significant interaction (F(1, 30) = 0.22, *p* = 0.64). Collectively, these results indicated that a single binge exposure impacts both antioxidant levels and membrane damage, so we proceeded to investigate outcomes following multiple binge exposures.

### 3.2. Multiple Binge

#### 3.2.1. Body Weight, Blood Ethanol, and Corticosterone

For body weight, one-way ANOVA was not significant in males (F(2, 26) = 2.95, *p* > 0.05) or females (F(2, 27) = 1.90, *p* > 0.05). For BEC, repeated measures ANOVA revealed a main effect of week (F(3, 84) = 2.97, *p* < 0.05); however, there were no significant post-hoc comparisons at any given timepoint. There also was no effect of sex and no significant interaction (Table 1). This is consistent with our prior findings that BEC does not differ by sex [41]. For plasma corticosterone, two-way ANOVA revealed a significant main effect of treatment (F(2, 50) = 3.502, *p* = 0.038) but not sex (F(1, 50) = 0.007, *p* = 0.93) and no significant interaction (F(2, 50) = 1.92, *p* = 0.15). To follow up on the significant treatment effect, we collapsed across sex. Tukey’s test revealed that binge alcohol elevated corticosterone compared to naïve animals (*p* = 0.030) but not controls (*p* = 0.34). Controls were not different from naïve (*p* = 0.45), indicating that gavage experience alone did not elevate corticosterone levels.

#### 3.2.2. Effect of Multiple Binge on Locomotor Activity, Anxiety-Like Behavior, and Behavioral Despair

The OF test was performed 48 h after the last binge treatment (Figure 3A,B). For distance traveled, two-way ANOVA revealed a significant main effect of sex (F(1, 53) = 36.07, *p* < 0.05) such that females traveled further than males. There was also a significant main effect of treatment (F(2, 53) = 6.36, *p* < 0.05), with binged animals of both sexes traveling the shortest distance. The interaction was not significant (F(2, 53) = 0.89, *p* > 0.05). Analysis of time spent in the center of the OF box revealed no main effects of sex (F(1, 52) = 2.76, *p* = 0.07) or treatment (F(2, 52) = 0.16, *p* = 0.68) and no interaction (F(2, 52)= 2.38, *p* = 0.1).

The EPM test was done following the OF, on the same day. Percent time (Figure 4A–C) was calculated to determine animals’ preference for each zone (closed arms, open arms, center). For % center duration, there was no significant main effect of sex (F(1, 53) = 0.46, *p* = 0.5) or treatment (F(2, 53) = 1.46, *p* = 0.24) and no significant interaction (F(2, 53) = 1.91, *p* > 0.05). In the open arms, we observed a main effect of sex (F(1, 53) = 11.09, *p* < 0.005) such that females spent more percent time there, but no effect was observed for treatment (F(2, 53) = 1.57, *p* = 2.16) and no significant interaction (F(2, 53) = 1.58, *p* > 0.05). For the closed arms, there was a significant main effect of sex (F(1, 53) = 8.21, *p* < 0.05) such that females spent less percent time there. There was also a significant main effect of treatment (F(2, 53) = 6.95, *p* < 0.05), with control animals of both sexes spending the most percent time there. The interaction was not significant (F(2, 53) = 0.66, *p* > 0.05). Head entries (Figure 4D,F) were tallied for all zones. For center zone entries, there was a significant main effect of sex (F(1, 53) = 14.89, *p* < 0.05), with females overall showing more. There was also a significant main effect of treatment (F(2, 53) = 3.80, *p* < 0.05), with binged animals of both sexes showing the fewest head entries in the center zone. The interaction was not significant (F(2, 53) = 0.80, *p* > 0.05). Analysis for open arm entries showed main effect of sex (F(1, 53) = 10.16, *p* < 0.05), again, with females showing more, but not treatment (F(2, 53) = 0.65, *p* > 0.05) and no significant interaction (F(2, 53) = 1.80, *p* > 0.05). Similarly, closed arm entries comparisons showed a main effect of sex (F(1, 53) = 17.22. *p* < 0.05), with females entering this area more, but no effect of treatment (F(2, 53) = 0.99, *p* > 0.05) and no significant interaction (F(2, 53) = 0.40, *p* > 0.05). Overall, these EPM results reflect only one effect of binge alcohol, namely reduced entries in the center zone, indicating anxiety-like effects in both sexes. These results also replicate the well-established sex difference in EPM, with females showing more overall activity and less anxiety-like behavior [55].

Training in the FST occurred the day after OF/EPM, and testing occurred the day after that (see Figure 1). For immobility time in the FST (Figure 3C), the main effect of sex was not significant (F(1, 53) = 2.047, *p* > 0.05), but the main effect of treatment was (F(2, 53) = 4.39, *p* < 0.05). The interaction was not significant (F(2, 53) = 1.19, *p* > 0.05). After collapsing across sex, Tukey’s test revealed that binged animals showed more immobility than control (*p* = 0.015) but not naïve (*p* = 0.16). Control and naïve were not different (*p* = 0.59). There were no significant differences among groups for vertical and horizontal swimming strategies.

#### 3.2.3. Effects of Repeated Binge on the Antioxidant Capacity of the Brain

The PFC plays a role in affect-related functions and is highly innervated from brain regions responsible for mood, learning, and memory [56]. To determine if the PFC is susceptible to changes in antioxidant activity from repeated binge alcohol exposure, we measured GSH levels and GR, GPx, and GST antioxidant enzyme activities (Figure 5). Percent values were calculated by dividing antioxidant concentrations from average naïve male group concentration for each antioxidant measure and discrete brain region. Analysis of glutathione levels in the PFC showed a significant main effect of treatment (F(2, 53) = 6.57, *p* < 0.05)). The main effect of sex fell just short of significance (F(1, 53) = 3.83, *p* = 0.056), and the interaction was not significant (F(2, 53) = 2.92, *p* = 0.062). Following up on the main effect of treatment, Tukey’s test showed binged animals overall had higher GSH levels compared to controls (*p* = 0.0048) and naïve (*p* = 0.020). Controls and naïve were not different (*p* = 0.88). For GR, there were no significant main effects of treatment (F(2, 53) = 0.21, *p* = 0.80) or sex (F(1, 53) = 2.93, *p* = 0.09) and no significant interaction (F(2, 53) = 3.01, *p* = 0.057). For GST, there was no significant main effect of treatment (F(2, 53) = 0.92, *p* = 0.39). There was a significant main effect of sex (F(1, 53) = 5.38, *p* = 0.02) such that females overall had higher levels than males. There was no significant interaction (F(2, 53) = 1.89, *p* = 0.16). For GPx, there were no significant main effects of treatment (F(2, 53) = 1.22, *p* = 0.30) or sex (F(1, 53) = 2.56, *p* = 0.12), but the interaction was significant (F(2, 53) = 3.49, *p* = 0.038). The only significant Tukey’s post hoc comparison was between the male and female binge groups (*p* = 0.029).

We assessed hippocampal GSH levels and antioxidant enzyme activities to determine if repeated binge alcohol treatment changed the antioxidant capacity in a brain region responsible for learning and memory (Figure 6). Analysis of GSH levels in the hippocampus showed a significant interaction (F(2, 53) = 6.79, *p* < 0.05) but no main effects of sex (F(1, 53) = 1.95, *p* = 0.17) or treatment (F(2, 53) = 0.48, *p* = 0.62). Tukey’s post-hoc tests revealed that male controls had higher GSH compared to male naïve (*p* = 0.044). For GR, there was no significant interaction (F(2, 53) = 0.82, *p* = 0.45) and no main effects of sex (F(1, 53) = 0.33, *p* = 0.56) or treatment (F(2, 53) = 0.87, *p* = 0.42). Additionally, for GST, there was no significant interaction (F(2, 53) = 1.85, *p* = 0.16) and no main effects of sex (F(1, 53) = 0.12, *p* = 0.72) or treatment (F(2, 53) = 0.87, *p* = 0.42). Finally, for GPx, there was no significant interaction (F(2, 53) = 1.10, *p* = 0.34) and no main effects of sex (F(1, 53) = 0.57, *p* = 0.39) or treatment (F(2, 53) = 0.06, *p* = 0.93).

The striatum plays a role in learned behaviors for drugs of abuse [57]. Therefore, we assessed GSH levels and glutathione-related antioxidant enzymes activities to determine if antioxidant competency changed with binge alcohol treatments (Figure 7). Analysis of GSH levels showed a significant interaction (F(2, 53) = 9.93, *p* < 0.05) but no main effects of sex (F (1, 53) = 1.19 *p* = 0.28) and treatment (F(2, 53) = 0.76, *p* = 0.47)). Tukey’s pos-hoc showed that control males were significantly different from male naïve and female control. Binge alcohol did not influence GR in the striatum, as there was not a significant interaction (F(2, 53) = 1.15, *p* = 0.32), and no significant main effects of sex (F(1, 53) = 2.49, *p* = 0.12) or treatment (F(2, 53) = 0.42, *p* = 0.65). The same was true for GST, with no significant interaction (F(2, 53) = 0.3, *p* = 0.74) and no main effects of sex (F(1, 53) = 0.01, *p* = 0.91) or treatment (F(2, 53) = 1.11, *p* = 0.33). For GPx in the striatum, there was no significant interaction (F(2, 53) = 1.13, *p* = 0.32), but there was a main effect of sex (F(1, 53) = 5.83, *p* < 0.05) such that females overall had higher levels. The main effect of treatment (F(2, 53) = 0.95, *p* = 0.39) was not significant.

#### 3.2.4. Effects of Repeated Binge on Brain Levels of Oxidative Damage

Lipid peroxidation was assessed by measuring malondialdehyde concentrations in the same brain regions processed for the antioxidant enzymes activities assays. We calculated percentages by dividing MDA concentrations from average male naïve group concentrations and multiplying by 100 (Table 2). In the PFC, there was no significant main effect of treatment (F(2, 53) = 2.73, *p* = 0.07) or sex (F(1, 53) = 1.12, *p* = 0.29) and no significant interaction (F(2, 53) = 0.38, *p* = 0.68). In the hippocampus, there were no significant main effects of treatment (F(2, 52) = 0.15, *p* = 0.85) or sex (F(1, 52) = 0.79, *p* = 0.37). The interaction was significant (F(2, 52) = 3.23, *p* < 0.05), but none of the Tukey’s post-hoc comparisons reached significance. In the striatum, there was a significant main effect of treatment (F(2, 53) = 3.75, *p* < 0.05) but not sex (F(1, 53) = 3.56, *p* = 0.06) and no significant interaction (F(2, 53) = 0.15, *p* = 0.85). Following up on the main effect of treatment, Tukey’s test showed binged animals overall had lower levels compared to naïve.

Protein oxidation was assessed using brain region samples processed for GSH levels. Percentages were calculated as previously discussed. In the PFC, there was no significant main effect of treatment (F(2, 51) = 1.46, *p* = 0.24). The main effect of sex was significant (F(1, 51) = 4.46, *p* < 0.05) such that females had overall higher levels. The interaction was not significant (F(2, 51) = 0.28, *p* = 0.75). In the hippocampus, there were no significant main effects of sex (F(1, 51) = 0.19, *p* = 0.66) or treatment (F(2, 51) = 0.20, *p* = 0.81) and no significant interaction (F(2, 51) = 1.24, *p* = 0.29). Finally, in the striatum, there were no significant main effects of sex (F(1, 51) = 0.24, *p* = 0.62) or treatment (F(2, 51) = 1.31, *p* = 0.27) and no significant interaction (F(2, 51) = 2.34, *p* = 0.10).

## 4. Discussion

In preclinical models, binge alcohol induces neuropathologies like synaptic degeneration, neuroinflammation, and behavior changes [5]. One proposed mechanism for these pathologies is the increased reactive oxygen species produced from alcohol metabolism [29]. Given that there is very little mechanistic evidence to support sex differences from binge alcohol in preclinical models, we set out to investigate affective changes, antioxidant activity, and oxidative stress differences from binge alcohol exposure in adult male and female Long–Evans rats. We predicted that female subjects would exhibit higher anxiety/depressive-like behaviors, increased oxidative stress and GSH levels, and decreased glutathione-related antioxidant enzymes.

These hypotheses were not supported. Although binge alcohol produced effects on behavior and oxidative stress parameters, they did not differ by sex. For behavior, we found that binged animals of both sexes showed anxiety- and depressive-like behaviors. Although binged animals overall traveled the shortest distance in the open field, binge exposure had no effect on time spent in the center. In the EPM, naïve animals overall made the most head entries into the center zone, and binged animals made comparably fewer. In general, on these two behavioral tests, we found that females, regardless of treatment, were significantly more explorative than males. This observation is typical for female behavior in OF and the EPM [56]. Finally, in the FST, we found that repeated binge exposure increased immobility time regardless of sex and independent of effects on horizontal or vertical swim strategies.

Anxiety is comorbid with binge drinking and has been linked to oxidative stress [36]. The tripeptide GSH is one of a few endogenous antioxidants in the brain that detoxifies neurons from oxidative damaging free radicals [19]. Xenobiotics, such as alcohol, can overwhelm antioxidant activity and lead to demonstrable signs of cellular stress, like lipid peroxidation and protein carbonylation [21]. This prolonged oxidative stress in the brain regions sensitive to alcohol may be one mechanism that induces cognitive impairment, mood disorders, and neurodegeneration [58]. Little is known, however, regarding whether acute and repeated binge alcohol exposures impact GSH levels and oxidative damage differently in male and female brains. In the present study, we tested the immediate effects of binge alcohol on GSH levels as well as oxidative damage in the PFC, hippocampus, and striatum of both sexes. We found that a single binge alcohol exposure elevated GSH levels in the PFC of both sexes and that this was associated with increased lipid peroxidation. In contrast, in the hippocampus, binge exposure decreased GSH levels, and this occurred in the absence of any detectable membrane damage. In the striatum, there were no changes in either GSH levels or membrane damage. Collectively, these results suggest that an acute binge alcohol exposure induces membrane damage and a compensatory increase in antioxidant level independent of sex but in a region-specific manner.

We next examined the effect of repeated binge exposure on antioxidant levels and membrane damage. Similar to our findings after a single exposure, four binge exposures increased GSH levels in the PFC regardless of sex. Unlike our single-binge findings, however, the increased GSH levels occurred in the absence of any detectable membrane damage, suggesting that elevated antioxidant levels successfully buffer the PFC against repeated binge exposure. There was a sex difference in the effect of binge on GPx levels, however, as binged males presented higher GPx levels compared to binged females. Since GPx catalyzes the reduction of various hydroperoxides to H_2_O via oxidation of reduced GSH into its disulfide form (GSSH), this increase in GPx activity in binged males could be a result of the glutathione redox cycle activation. Both sexes showed significant antioxidant response in the PFC after repeated exposure, and both sexes showed behavioral changes consistent with the contribution of the PFC and its circuitry to depressive- and anxiety-like behavior, which preclinical studies using optogenetic/chemogenetic techniques have begun to elucidate (for review, see [56]). Neither sex showed binge-induced antioxidant changes or membrane damage in the hippocampus after repeated exposure. In the striatum, only control males showed increased GSH levels. Females, on the other hand, showed neither antioxidant changes nor membrane damage in the striatum. It is possible that other mechanisms outside of increased GSH protected the female striatum, such as increased activity of anti-inflammatory cytokines. It is important to emphasize that GSH level can modulate neural activity and link it to mPFC and behavior. Maas and colleagues, using apomorphine-susceptible (APO-SUS) rat model to study schizophrenia, reported a central role for GSH antioxidant metabolism. They found that treatment with the GSH precursor *N*-acetylcysteine (NAC) restored not only GSH metabolism but also improved mPFC hypomyelination and cognitive functioning of APO-SUS rats [59].

Lipid peroxidation is one sign of oxidative damage that arises in the brain due to excessive alcohol consumption [19,21,60]. This pattern of toxicity in neurons can promote a pro-inflammatory environment that leads to neuronal loss and behavioral deficits [36]. The severity of these pathologies is not well characterized in female subjects, who are more likely to suffer from mood disorders and memory deficits with excessive alcohol consumption. Here, measurements of TBARS revealed that lipid peroxidation was significantly higher in the PFC of both male and female binge subjects after a single (but not repeated) binge exposure. Our lipid peroxidation findings are in agreement with a previous study that showed increased TBARS levels with a single binge dose but no changes with chronic binge exposure in male rats [61]. However, that study also observed decreased GSH levels in total brain homogenates with acute exposure, suggesting that GSH metabolism may vary with brain region, whereas lipid peroxidation is similarly produced throughout the brain in response to acute and chronic binge alcohol. Since increased lipid peroxidation products have been related to decreased neuronal viability [62,63], these enhanced MDA levels in the PFC may systematically contribute to brain damage, leading to acute behavioral alterations and chronic neurodegeneration. In addition, alcohol-induced lipid peroxidation by oxidative stress and its products decrease the intracellular reduced GSH and increase its oxidized form [64]. Here, we measured the total GSH levels, so we cannot discard the possibility that the oxidized form is responsible for this increase in GSH levels in the PFC of binged animals. It is interesting that we found no lipid peroxidation or protein oxidation with repeated binge exposure, yet affective changes were present, indicating that other mechanisms could be influencing those behaviors, such as reactive nitrogen species or neuroinflammation.

The response of neurons to oxidative stress is not uniform in the brain. While many brain neurons can better tolerate oxidative stress, there are some brain regions that are more vulnerable. In this study, the majority of our findings on antioxidant changes and membrane damage were specific to the PFC, which is highly susceptible to alcohol-induced damage [65]. This is particularly important because the inability to abstain from alcohol can be linked with PFC malfunctions, including executive dysfunctions, such as deficits in working memory, impulse control, and decision making [66]. In the present study, we did not observe increased binge-induced membrane damage in the hippocampus and striatum; in fact, female binge subjects showed decreased lipid peroxidation relative to controls, suggesting region-dependent susceptibility. This is consistent with a previous study showing brain region-dependence of markers of oxidative stress after varying durations of chronic alcohol exposure [67]. However, recent studies have suggested that many other brain regions are involved in alcohol intake and the negative effects of withdrawal. Accumulating evidence suggests that the lateral habenula (LHb), an epithalamic structure that connects the forebrain with the midbrain, may play a crucial role [68,69]. Several preclinical studies suggest that acute and repeated alcohol exposure increases the strength of excitatory synapses and excitability of LHb neurons, which is concomitant with the affective psychiatric behaviors occurring during alcohol withdrawal [70] Besides providing reducing equivalents for maintaining oxidant homeostasis, GSH plays a role in intra- and intercellular signaling in the brain [71]. Therefore, it is possible that GSH levels in the LHb could modulate alcohol consumption and related psychiatric disorders.

We included naïve (cage control) animals in our experiments in order to control for potential effects of intragastric gavage. Previously, we have shown that gavage with isocaloric control diet did not impact cellular outcome measures after repeated binge exposure [40]. In the present study, we observed several effects of gavage experience in our antioxidant assays. In males, gavage increased GSH levels compared to naïve in the hippocampus and striatum. This is not likely due to stress, as gavage experience alone did not increase corticosterone, consistent with our prior evidence that gavage experience is not detectably stressful [41]. Although the control diet is isocaloric with the alcohol diet, the calories are derived from refined sugar (rather than alcohol). It is possible that this is capable of influencing antioxidant levels [72].

Blood alcohol levels that exceed 80 mg/dL within 2 h is considered a binge episode, though these levels could 2–3 times higher in the average American binge drinker, who consumes seven drinks per binge [73]. Here, we report equivalent BECs between male and female groups each week confirming that both groups received similar alcohol treatments regardless of weight and consistent with our prior findings showing similar BEC between the sexes [41]. Moreover, the BEC are substantial (mean 176 mg/dL) and consistent with those linked to brain damage and neurodegeneration [74]. The heavy consumption of alcohol has been shown to alter the levels of the stress hormone, corticosterone. In the present study, the fourth binge alcohol exposure significantly raised corticosterone levels, consistent with the idea that alcohol represents a physiological stressor [75].

## 5. Conclusions

To our knowledge, this is the first investigation of sex-dependent effects of binge alcohol on affective behaviors and brain antioxidant and oxidative stress levels. Overall, we found little evidence for sex differences in the effects of either acute or repeated binge alcohol, as we observed similar behavioral and antioxidant changes in both sexes. Future studies are necessary to determine whether antioxidant modulation protects against binge-induced anxiety/depression-like behaviors in males and females.

## Figures and Tables

**Figure 1 brainsci-11-01250-f001:**
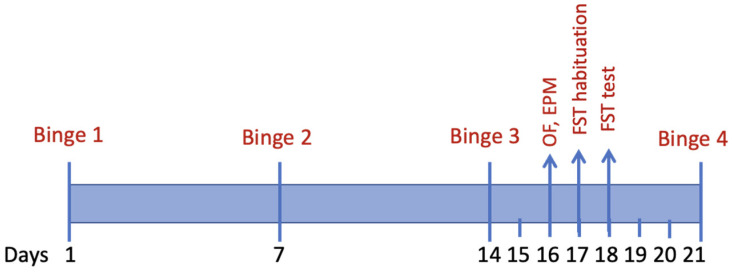
Timeline of experimental manipulations. Forty-eight hours after the third binge, rats were behaviorally tested for the next three days.

**Figure 2 brainsci-11-01250-f002:**
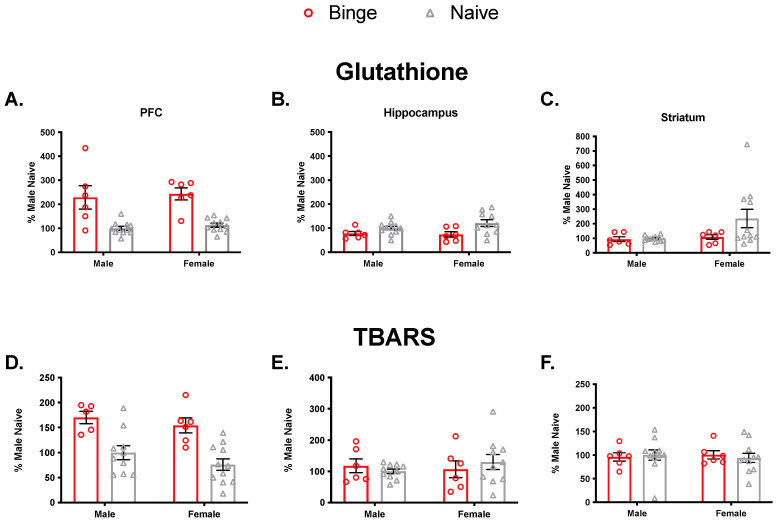
Effects of acute binge on glutathione levels and lipid peroxidation. A single binge significantly elevated GSH level in PFC (**A**), reduced it in hippocampus (**B**), and did not change levels in the striatum (**C**). Bars represent ± SEM from 6–11 subjects per group. Significant lipid peroxidation was detectable in both male and female PFC compared to naïve subjects (**D**). There was no detectable lipid peroxidation damage from a single binge treatment in hippocampus (**E**) or striatum (**F**).

**Figure 3 brainsci-11-01250-f003:**
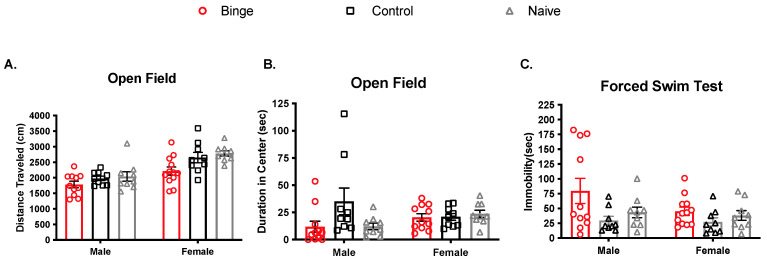
Effect of multiple binge on locomotor activity and behavioral despair in young adult rats. OF data was collected and analyzed from automatic tracking software in the first five minutes. (**A**) There was a main effect of sex, with females traveling further than males. (**B**) Time spent in the center of the OF did not differ by sex or binge exposure. (**C**) In the forced swim test, binged animals showed more immobility time than controls. Bars represent ± SEM from 9–12 subjects per group.

**Figure 4 brainsci-11-01250-f004:**
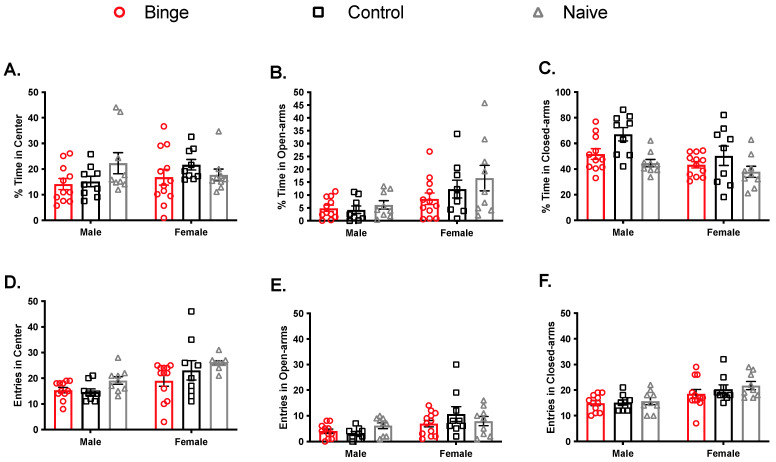
Effect of repeated binge exposure on anxiety-like behavior in males and females. Although there were no significant differences in % time in the center (**A**), binged animals of both sexes made the fewest head entries into the center (**D**). Females spent more % time (**B**) and made more head entries into the open arms (**E**), but there were no effects of binge. For the closed arms, males spent more time there compared to females (**C**). Females made more head entries into the closed arms (**F**) compared to males. Bars represent ± SEM from 9–12 subjects per group.

**Figure 5 brainsci-11-01250-f005:**
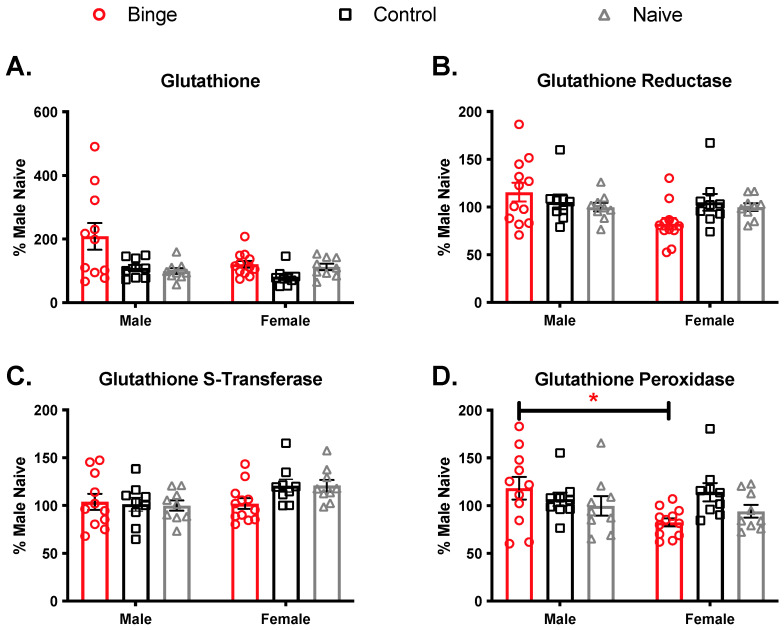
Effects of multiple binge in antioxidant capacity in the prefrontal cortex Binge alcohol treatments increased glutathione (GSH) levels overall in the PFC (**A**). No significant effects were observed in PFC samples for (**B**) Glutathione reductase. There was a main effect of sex for Glutathione S-Transferase (**C**), with females having overall higher levels. For glutathione peroxidase (**D**), binged males had higher levels than binged females. Bars depict ± SEM from percent naïve in (9–12) animals. * *p* < 0.05.

**Figure 6 brainsci-11-01250-f006:**
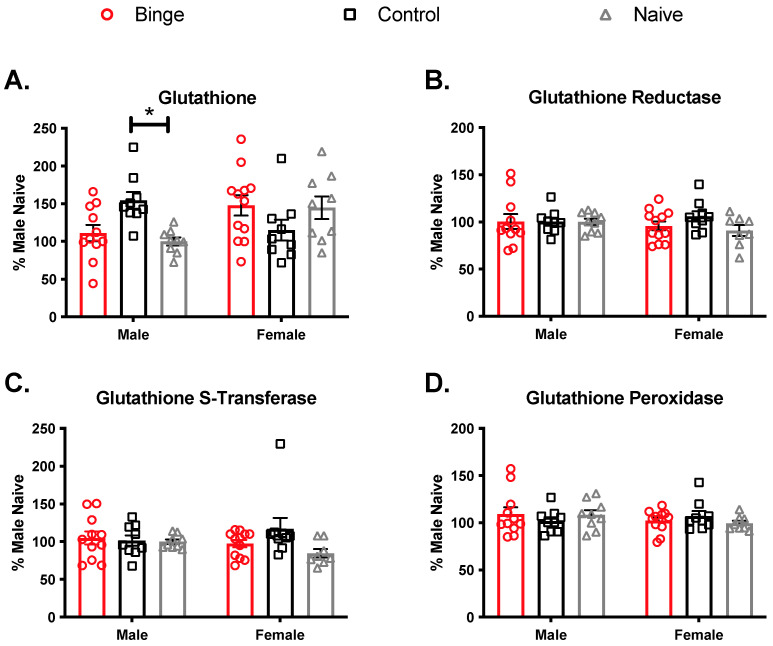
Effects of multiple binge on antioxidant competence in the hippocampus. Elevated GSH level was detected in hippocampal samples of males treated with control diet compared to naïve (**A**). There were no significant of sex or binge treatment in (**B**) Glutathione reductase, (**C**) Glutathione S-Transferase, or (**D**) Glutathione Peroxidase assays activities. Bars depict ± SEM from percent naïve in (9–12) animals. * *p* < 0.05.

**Figure 7 brainsci-11-01250-f007:**
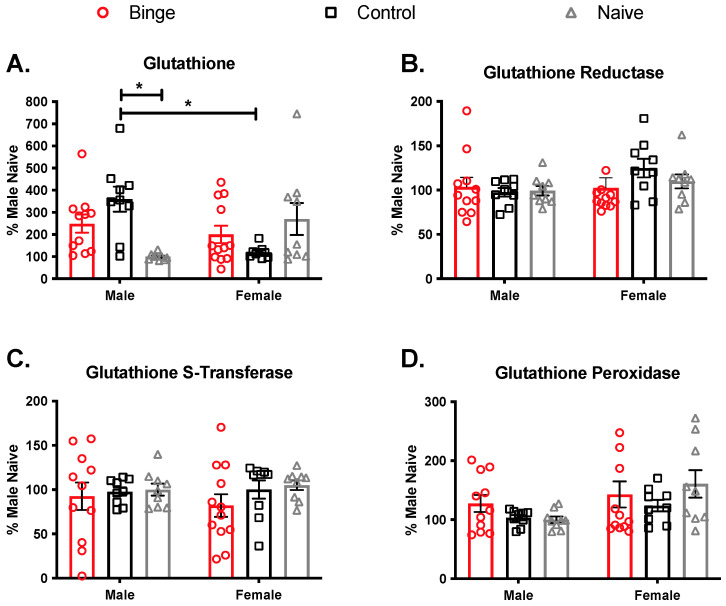
Effects of multiple binge on antioxidant capacity in the striatum. Gavage experience significantly increased glutathione levels in males compared to male naïve and female controls (**A**). There was no effect of binge alcohol on glutathione reductase (**B**) or Glutathione S-Transferase (**C**). For glutathione peroxidase (**D**), there was a main effect of sex such that females had overall higher levels compared to males. Bars depict ± SEM from percent male naïve in (9–12) animals. * *p* < 0.05.

**Table 1 brainsci-11-01250-t001:** Blood ethanol concentrations (BEC) in binged groups across time and corticosterone levels in all groups after final binge exposure.

**BEC (mg/dL)**	**Binge 1**	**Binge 2**	**Binge 3**	**Binge 4**
male	199.6 ± 5.2	141.7 ± 17.9	172.0 ± 8.4	162.4 ± 18.2
female	198.1 ± 17.4	157.8 ± 5.7	151.2 ± 14.1	228.1 ± 13.2
**Corticosterone (ng/mL)**	**Binge**	**Control**	**Naive**	
male	70.3 ± 7.8	75.7 ± 7.6	54.5 ± 15.2	
female	95.7 ± 20.3	58.1 ± 6.6	49.3 ± 5.8	

**Table 2 brainsci-11-01250-t002:** Measures of lipid peroxidation (TBARS) and protein damage (protein carbonyls) in brain regions of interest in binged and control groups. Data are expressed as % naïve male.

TBARS	PFC		Hippocampus		Striatum	
	Binge	Control	Binge	Control	Binge	Control
Male Female	138.3 ± 27.3 119.7 ± 17.2	107.2 ± 22.1 108.6 ± 22.0	121.9 ± 9.4 106.6 ± 9.8	152.0 ± 30.1 93.3 + 15.3	73.3 ± 14.4 47.7 + 10.1	83.8 ± 13.3 64.1 ± 12.2
**Protein Carbonyls** Male Female	103.7 ± 8.8 117.0 ± 7.2	86.2 ± 8.8 107.4 ± 7.5	111.3 ± 8.9 104.3 ± 6.4	113.3 ± 3.4 111.6 ± 6.0	106.81 ± 5.1 91.0 ± 7.9	100.3 ± 6.5 90.6 ± 6.0

## Data Availability

Study data are available by e mailing either corresponding author.

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
