# Peer review of "Changes in Affective Behavior and Oxidative Stress after Binge Alcohol in Male and Female Rats"

_brainsci, 2021, doi:10.3390/brainsci11091250_

Round 1

Reviewer 1 Report

This article by Cortez et al, “Sex-specific Changes in Affective Behavior and Oxidative Stress After Binge Alcohol in Rats,” describes the effects of repeated intra-gastric alcohol administrations on measures of depression and anxiety-like behavior, glutathione levels and function of its metabolic enzymes, and lipid peroxidation in male and female rats. The main findings reported are that withdrawal from a binge” alcohol exposure regimen results in sex-specific increases in anxiety- and depression-like behavior, as well as sex and brain-region specific changes in glutathione levels.  In general, this paper is clearly written and the results are interesting.  However, there are several major concerns that need to be addressed. 

Major Concerns

  1. The presentation of the statistical results in the Results section is inconsistent across experiments, omits results from multiple analyses, and in several cases, includes inappropriate post hoc tests. The behavioral data, glutathione levels, and enzymatic activity were analyzed by 2-way ANOVA, with each analysis having the potential to detect a main effect of treatment, a main effect of sex, and a treatment by sex interaction.  Only a handful of descriptions of the dependent measures mention the main effects and interactions, and in several cases, post hoc analyses were conducted inappropriately in the absence of a statistical interaction, as well as unnecessarily where there was a main effect of sex, or in the main effect of treatment described for the single exposure results presented in Figure 2.  This makes understanding and interpreting the differences among conditions and sex very difficult.  The results section needs to be rewritten to clarify the statistical results. 

  1. The description of the behavioral testing methods is missing important details. The timeline of behavioral testing needs to be included, and a statement as to the potential for carry over effects across tests described. In addition, more details of the behavioral testing, including the lighting conditions, the time of day that testing occurred, acclimation conditions, etc, need to be included in the Methods section.   Further , more details about how the behavioral data were calculated need to be presented.  For the open field, was velocity calculated by dividing the total amount of activity by the duration of the session?  If so, these data (3B) are redundant with those in 3A and can be eliminated.  Similarly, how were the percent times calculated for the EPM?  Finally, how were head entries calculated in the EPM?  Given that automated tracking was used, such that the rat’s location was approximated by the location of its front shoulder, it’s not clear how a head entry into an arm was discriminated from a whole body entry. 

  1. More details of the ROUT procedure for the detection of outliers (p 5) need to be provided, and a statement added that says how many data points from which analyses were eliminated.

  1. Given that one of the goals of the experiments was to relate GSH levels and metabolism to the behavioral consequences of withdrawal from a binge alcohol procedure, the finding of correlations between the GSH biochemical measures and behavioral measures would be very useful. Do the behavioral results from the forced swim test, EPM or open field correlate with these measures?

  1. The formatting for Table 2 is off, and a legend is needed.

Minor Concerns

  1. This sentence in the introduction (Lines 78-80) is confusing and/or inaccurate: “[u]sing genetic mouse models, Hovatta l., et al., 2005, discovered that inhibiting expression of oxidative stress genes in anxious inbred mice, prevented anxiety in a non-anxious mouse model expressing this same set of 80 genes [35].”

  1. This sentence (line 245-246): “[t]here was no significant lipid peroxidation observed for the hippocampus or striatum samples” needs to be revised to include the phrase “no differences.”

  1. The results for the analysis of the BEC (beginning on line 262) states that there was a main effect of week, but no effect of sex, which was interpreted as being consistent with prior findings that BEC does not differ by sex or change over the course of weekly binges. This cannot be the case, as there was a main effect of time that was found in the analysis. This sentence should be revised or eliminated. 

  1. The analysis of corticosterone levels found a main effect of treatment but not sex, and no interaction was reported. In the absence of significant interaction, the post hocs comparing treatment effects within males and females are not justified, and the conclusion that females show a stronger stress response unwarranted.

  1. One explanation for a lack of a sex effect (p 433-436) is that binge alcohol “may have an effect on gonadal hormones and depress females natural tendency to explore new environments [59].: In the absence of any further explanation – and particularly, given that this explanation is in regards to a null result – this seems to be an overinterpretation.

Author Response

REVIEWER 1

This article by Cortez et al, “Sex-specific Changes in Affective Behavior and Oxidative Stress After Binge Alcohol in Rats,” describes the effects of repeated intra-gastric alcohol administrations on measures of depression and anxiety-like behavior, glutathione levels and function of its metabolic enzymes, and lipid peroxidation in male and female rats. The main findings reported are that withdrawal from a binge” alcohol exposure regimen results in sex-specific increases in anxiety- and depression-like behavior, as well as sex and brain-region specific changes in glutathione levels.  In general, this paper is clearly written and the results are interesting.  However, there are several major concerns that need to be addressed. 

Major Concerns

  1. The presentation of the statistical results in the Results section is inconsistent across experiments, omits results from multiple analyses, and in several cases, includes inappropriate post hoc tests. The behavioral data, glutathione levels, and enzymatic activity were analyzed by 2-way ANOVA, with each analysis having the potential to detect a main effect of treatment, a main effect of sex, and a treatment by sex interaction.  Only a handful of descriptions of the dependent measures mention the main effects and interactions, and in several cases, post hoc analyses were conducted inappropriately in the absence of a statistical interaction, as well as unnecessarily where there was a main effect of sex, or in the main effect of treatment described for the single exposure results presented in Figure 2.  This makes understanding and interpreting the differences among conditions and sex very difficult.  The results section needs to be rewritten to clarify the statistical results. 

Response: It was necessary to re-run the statistical analyses to address the concerns of R2. We have thus re-written the Results section, including all F and p values, regardless of significance, and avoided between-sex comparisons in the absence of an interaction.

  1. The description of the behavioral testing methods is missing important details. The timeline of behavioral testing needs to be included, and a statement as to the potential for carry over effects across tests described. In addition, more details of the behavioral testing, including the lighting conditions, the time of day that testing occurred, acclimation conditions, etc, need to be included in the Methods section.   Further , more details about how the behavioral data were calculated need to be presented.  For the open field, was velocity calculated by dividing the total amount of activity by the duration of the session?  If so, these data (3B) are redundant with those in 3A and can be eliminated.  Similarly, how were the percent times calculated for the EPM?  Finally, how were head entries calculated in the EPM?  Given that automated tracking was used, such that the rat’s location was approximated by the location of its front shoulder, it’s not clear how a head entry into an arm was discriminated from a whole body entry. 

Response: In the Methods section of our revised manuscript, we have addressed the potential for stress carry-over effects across behavioral tests, and added details regarding behavioral testing.

  1. More details of the ROUT procedure for the detection of outliers (p 5) need to be provided, and a statement added that says how many data points from which analyses were eliminated.

Response: In the revised manuscript, we have specified that the ROUT outlier test was used to identify and remove a single data point from the binge female group in the corticosterone analysis, which had a more than two-fold higher value than the next highest. We have also detailed cases in which there was insufficient tissue available to run all biochemical analyses for a given animal – this information is found in section 2.5 Tissue Preparation.

  1. Given that one of the goals of the experiments was to relate GSH levels and metabolism to the behavioral consequences of withdrawal from a binge alcohol procedure, the finding of correlations between the GSH biochemical measures and behavioral measures would be very useful. Do the behavioral results from the forced swim test, EPM or open field correlate with these measures?

Response: This is a great suggestion. But, we ran correlations between antioxidant levels and behavioral outputs, and unfortunately, the only significant correlation was between PFC GSH and immobility in the FST in binged males. We have not included this in the revised manuscript because it was not significant if all males were included.

  1. The formatting for Table 2 is off, and a legend is needed.

Response: Brain Science’s editorial office formatted our tables and their formatting is in our revised manuscript. We have added a legend to Table 2.

Minor Concerns

  1. This sentence in the introduction (Lines 78-80) is confusing and/or inaccurate: “[u]sing genetic mouse models, Hovatta l., et al., 2005, discovered that inhibiting expression of oxidative stress genes in anxious inbred mice, prevented anxiety in a non-anxious mouse model expressing this same set of 80 genes [35].”

Response: We thank the reviewer for pointing this out. Please see that now this sentence was modified: “Using different inbred strains of mice and lentivirus-mediated gene transfer to overexpress glyoxalase 1 and glutathione reductase 1 genes in the mouse brain, Hovatta and colleagues found that these genes have a causal role in the genesis of anxiety. Both of these genes are involved in oxidative stress metabolism, linking this pathway with anxiety-related behavior.”

  1. This sentence (line 245-246): “[t]here was no significant lipid peroxidation observed for the hippocampus or striatum samples” needs to be revised to include the phrase “no differences.”

Response: Please see that this sentence was eliminated when the Results section was rewritten according to the comments of R2.

  1. The results for the analysis of the BEC (beginning on line 262) states that there was a main effect of week, but no effect of sex, which was interpreted as being consistent with prior findings that BEC does not differ by sex or change over the course of weekly binges. This cannot be the case, as there was a main effect of time that was found in the analysis. This sentence should be revised or eliminated. 

Response: We thank the reviewer for pointing it out. Please see that now this sentence was revised.

  1. The analysis of corticosterone levels found a main effect of treatment but not sex, and no interaction was reported. In the absence of significant interaction, the post hocs comparing treatment effects within males and females are not justified, and the conclusion that females show a stronger stress response unwarranted.

Response: The post hoc comparisons were conducted within each sex, because of the main

effect of Treatment and the 3 conditions. In a two-factor study, the basic principles of statistical analysis allows for comparison among treatment means when the two factors do not interact (Wei J., et al., 2012)

  1. One explanation for a lack of a sex effect (p 433-436) is that binge alcohol “may have an effect on gonadal hormones and depress females natural tendency to explore new environments [59].: In the absence of any further explanation – and particularly, given that this explanation is in regards to a null result – this seems to be an overinterpretation.

Response: Please see that we have amended our coverage of this point in the Discussion.

Reviewer 2 Report

In this paper, Cortez et al. have described the sexually dimorphic nature of binge alcohol consumption in rats using behavior and oxidative stress readouts. The authors experimented with male and female rats with a weekly binge alcohol paradigm for four weeks and assessed anxiety and depressive-like behavior along with endpoint quantification of oxidative stress markers in different brain regions. Though the paper shows some sex-specific changes, the authors need to address major concerns to support their claims.

Major points:

  1. The introduction is missing the rationale for the author's prediction or hypothesis.
  2. A statistician should verify statistics. All the biochemistry assay results are normalized to a particular sex's naïve condition. If that is the case then, a two-way ANOVA for sex factor cannot be conducted as the normalization changes the baseline for each sex. Given that some of the manuscript's conclusions regarding sexual dimorphic behavior are based on these results, verification of these results is crucial.
  3. The authors have used two-month Long-Evans rats, the rationale for using young rats and the epidemiologically how that relates to humans should be included in the paper.
  4. Figure 1 needs to be revised to include the days when each behavior was conducted in week 3. According to the figure, it seems like all the behavior was conducted on the same day.
  5. Authors have elaborated on statistics in their results write-up but missed marking the stats on the figure. All the figures should at least have all significant stats, including two-way ANOVA results.
  6. Bodyweight data should be included.
  7. OF test is used by the authors to compute the locomotion of rats. Still, the same test can be mined to assess anxiety by calculating the time spent in zones– highly recommended as the rationale of the manuscript is to evaluate anxiety-like behavior due to alcohol binge. Check Jove article by Seibenhener and Wooten; 2015. (PMID: 25742564)
  8. The conclusion drawn by the authors that multiple binges in females cause anxiety-like behavior is based on EPM data where females show a significant decrease in entries compared to naïve, but the actual comparison should be made with the control group. This should be looked into more carefully.  
  9. The rationale of choosing different brain regions can be addressed clearly as well as what it means to have changes in one region but no difference in another.

Minor points:

  1. Did the authors use both the hemisphere for the analysis?
  2. Subheadings starting from the result section should be reformatted for appropriate spacing and punctuations.For example – 3. .Results --> 3. Results and 3. 1Single Binge --> 3.1 Single Binge
  3. Aesthetic input – The distinction of assays in a figure would help readers align themselves better while reading the manuscript. For example, in figure 3: A vertical or horizontal inclusion of test name in the figure would help distinguish data – OF for Fig. 3A, B, and FST for Fig. 3C-E.

Author Response

REVIEWER 2

In this paper, Cortez et al. have described the sexually dimorphic nature of binge alcohol consumption in rats using behavior and oxidative stress readouts. The authors experimented with male and female rats with a weekly binge alcohol paradigm for four weeks and assessed anxiety and depressive-like behavior along with endpoint quantification of oxidative stress markers in different brain regions. Though the paper shows some sex-specific changes, the authors need to address major concerns to support their claims.

Major points:

  1. The introduction is missing the rationale for the author's prediction or hypothesis.

Response: Please see that in the second paragraph in the introduction we stated the rationale for our hypothesis: ... “females are more susceptible to present acute and long-term alterations of mood and memory [17], and morphological changes [11] after binge drinking compared to males. Beyond that, oxidative balance also seems to be different between males and females. For example, Jung and Metzger [18] found oxidative stress sex difference after alcohol exposure.” Therefore, we predicted that female subjects would exhibit higher anxiety/depressive-like behaviors and increased oxidative stress.

  1. A statistician should verify statistics. All the biochemistry assay results are normalized to a particular sex's naïve condition. If that is the case then, a two-way ANOVA for sex factor cannot be conducted as the normalization changes the baseline for each sex. Given that some of the manuscript's conclusions regarding sexual dimorphic behavior are based on these results, verification of these results is crucial.

Response: We thank the reviewer for pointing out this error. We have now normalized all values in relation to male naïve. We have revised the Results and Discussion sections accordingly.

  1. The authors have used two-month Long-Evans rats, the rationale for using young rats and the epidemiologically how that relates to humans should be included in the paper.

Response: We used young adult rats in this study to mimic the human behavior where binge drinking is most common among younger adults aged 18–34 years (Krieger et al., 2018). Now this information was added to the Material and Methods.

  1. Figure 1 needs to be revised to include the days when each behavior was conducted in week 3. According to the figure, it seems like all the behavior was conducted on the same day.

Response: Please see that Figure 1 and it’s caption have been revised to enhance clarity.

  1. Authors have elaborated on statistics in their results write-up but missed marking the stats on the figure. All the figures should at least have all significant stats, including two-way ANOVA results.

Response: In our revised figures, all significant post hoc differences have been indicated. To avoid confusion in the graphs, we have not indicated the significant main effects in the graphs themselves, but those effects are noted in the figure captions.

  1. Bodyweight data should be included.

Response: We have now included body weight in Results section 3.2.1.

  1. OF test is used by the authors to compute the locomotion of rats. Still, the same test can be mined to assess anxiety by calculating the time spent in zones– highly recommended as the rationale of the manuscript is to evaluate anxiety-like behavior due to alcohol binge. Check Jove article by Seibenhener and Wooten; 2015. (PMID: 25742564)

Response: We appreciate that OF is sometimes used to gauge anxiety-like behavior in rodents. We believe EPM is a more sensitive measure. However, to address the reviewer’s concern, we, have added the data for and a graph (Figure 3B) of center duration in the open field.

  1. The conclusion drawn by the authors that multiple binges in females cause anxiety-like behavior is based on EPM data where females show a significant decrease in entries compared to naïve, but the actual comparison should be made with the control group. This should be looked into more carefully.  

Response: We clarify this point in our discussion and added a sentence stating that the stress caused by gavage alone was not sufficient to increase anxiety-like behavior.

  1. The rationale of choosing different brain regions can be addressed clearly as well as what it means to have changes in one region but no difference in another.

Response: Please see that this paragraph has been amended in the Discussion:

“The response of neurons to oxidative stress is not uniform in the brain. While many brain neurons can better tolerate oxidative stress, there are some brain regions that are more vulnerable. In this study the majority of our findings on antioxidant changes and membrane damage were specific to the PFC, which is highly susceptible to alcohol-induced damage [70]. This is particularly important, because the inability to abstain from alcohol can be linked with PFC malfunctions, including executive dysfunctions such as deficits in working memory, impulse control and decision making [71]. In the present study, we did not observe increased binge-induced membrane damage in the hippocampus and striatum, in fact, female binge subjects showed decreased lipid peroxidation, relative to controls, suggesting region-dependent susceptibility. This is consistent with a previous study showing brain region-dependence of markers of oxidative stress after varying durations of chronic alcohol exposure [72].”

Minor points:

  1. Did the authors use both the hemisphere for the analysis?

Response: Yes, we used both hemispheres. Please see that now this information was added to the text.

  1. Subheadings starting from the result section should be reformatted for appropriate spacing and punctuations.For example – 3. .Results --> 3. Results and 3. 1Single Binge --> 3.1 Single Binge

Response: Brain Science’s editorial office formatted our headings and subheadings and their formatting is shown in our revised manuscript.

  1. Aesthetic input – The distinction of assays in a figure would help readers align themselves better while reading the manuscript. For example, in figure 3: A vertical or horizontal inclusion of test name in the figure would help distinguish data – OF for Fig. 3A, B, and FST for Fig. 3C-E.

Response: We have included titles on the revised figures, to enhance clarity

Reviewer 3 Report

This is a very interesting study illustrating the role of glutathione in the mPFC during alcohol binge drinking in male and female mice. Overall, the findings are interesting and the methods are sounding. I only have a few minor comments.

  1. It would be great to discuss how glutathione level or ROS in general can affect neural activity and link it to mPFC and the behavior.
  2. Also, recent studies have suggested that many other brain regions are involved in alcohol intake and the negative effects of withdrawal (mostly see recent works from Thomas Jhou and Manuel Mameli). It would be interesting to also discuss how glutathione can affect the brain circuits in general.

Author Response

REVIEWER 3

This is a very interesting study illustrating the role of glutathione in the mPFC during alcohol binge drinking in male and female mice. Overall, the findings are interesting and the methods are sounding. I only have a few minor comments.

  1. It would be great to discuss how glutathione level or ROS in general can affect neural activity and link it to mPFC and the behavior.

Response: We thank the reviewer for his/her suggestion. In the revised manuscript, we have included this topic to the discussion.

It is important to emphasize that glutathione level can modulate neural activity and link it to mPFC and the behavior. Maas and colleagues, using apomorphine-susceptible (APO-SUS) rat model to study schizophrenia, reported a central role for GSH antioxidant metabolism. They found that treatment with the GSH precursor N-acetylcysteine (NAC) restored not only GSH metabolism, but also improved mPFC hypomyelination and cognitive functioning of APO-SUS rats (Neuropsychopharmacology (2021) 46:1161–1171).”

  1. Also, recent studies have suggested that many other brain regions are involved in alcohol intake and the negative effects of withdrawal (mostly see recent works from Thomas Jhou and Manuel Mameli). It would be interesting to also discuss how glutathione can affect the brain circuits in general.

Response: Please see the following paragraph was included in the discussion:

“However, recent studies have suggested that many other brain regions are involved in alcohol intake and the negative effects of withdrawal. Accumulating evidence suggests that the lateral habenula (LHb), an epithalamic structure that connects the forebrain with the midbrain may play a crucial role [72,73]. Several preclinical studies suggest that acute and repeated alcohol exposure increases the strength of excitatory synapses and excitability of LHb neurons, which is concomitant with the affective psychiatric behaviors occurring during alcohol withdrawal [74]. Besides providing reducing equivalents for maintaining oxidant homeostasis, GSH plays a role in intra- and intercellular signaling in the brain [75]. So, it is possible that GSH levels in the LHb could modulate alcohol consumption and ailments psychiatric disorders-related.”

Round 2

Reviewer 1 Report

The revisions to this manuscript, “Sex-specific Changes in Affective Behavior and Oxidative

Stress After Binge Alcohol in Rats,” have addressed most of the concerns I had with the original submission.  However, the presentation of statistical results for the behavioral experiments still needs to be improved. 

The authors note in their response to my first concern (Major Concern 1) that the revised manuscript does not include post hocs tests as follows up to non-statistically significant interactions.  This is true for some, but not all of the biochemical analyses, and not for the behavioral analyses or the corticosterone and body weight analyses.  Some of these analyses report only significance levels for main effects or the interactions, yet most (still) report post hoc comparisons.  These need to be corrected.  As a side note, the figure depicting velocity in the open field has been removed from Figure 3 (which eliminates a redundant analysis generated by dividing a data set by a constant) but the analysis of velocity is still presented in the text.  This should be removed.

The authors note in their response to minor concern 4 that Wei et al (2012) outlines a rationale for conducting post hocs in a two-way factorial design following the finding of a non-significant interaction term.  I can explicate why this rationale – although published – is incorrect (in short: main effects in a linear model are calculated for aggregate means, not individual cell means, as described in the paper), but instead, will just note that Wei et al (2012) proposes (in the text describing Table 1) that post hocs might be used following a two-way factorial analysis only if both main effects are significant. 

Part of the issue might be definitional/semantic: what the authors are presenting as post hoc tests could be regarded as “planned comparisons, " especially if there is a clearly stated reason for expecting differences among one or more of the groups.  Since this paper set out to examine sex differences in biochemical and behavioral responses to a binge-like ethanol administration procedure – and in the Introduction describes sex differences in alcohol-related responses - this seems like the necessary frame of reference from which to examine differences among cell means that were not large enough to drive a statistical interaction: planned comparisons.  The results should be presented as such. 

Author Response

REVIEWER 1

The revisions to this manuscript, “Sex-specific Changes in Affective Behavior and Oxidative Stress After Binge Alcohol in Rats,” have addressed most of the concerns I had with the original submission.  However, the presentation of statistical results for the behavioral experiments still needs to be improved. 

The authors note in their response to my first concern (Major Concern 1) that the revised manuscript does not include post hocs tests as follows up to non-statistically significant interactions.  This is true for some, but not all of the biochemical analyses, and not for the behavioral analyses or the corticosterone and body weight analyses.  Some of these analyses report only significance levels for main effects or the interactions, yet most (still) report post hoc comparisons.  These need to be corrected.  As a side note, the figure depicting velocity in the open field has been removed from Figure 3 (which eliminates a redundant analysis generated by dividing a data set by a constant) but the analysis of velocity is still presented in the text.  This should be removed.

The authors note in their response to minor concern 4 that Wei et al (2012) outlines a rationale for conducting post hocs in a two-way factorial design following the finding of a non-significant interaction term.  I can explicate why this rationale – although published – is incorrect (in short: main effects in a linear model are calculated for aggregate means, not individual cell means, as described in the paper), but instead, will just note that Wei et al (2012) proposes (in the text describing Table 1) that post hocs might be used following a two-way factorial analysis only if both main effects are significant. 

Part of the issue might be definitional/semantic: what the authors are presenting as post hoc tests could be regarded as “planned comparisons, " especially if there is a clearly stated reason for expecting differences among one or more of the groups.  Since this paper set out to examine sex differences in biochemical and behavioral responses to a binge-like ethanol administration procedure – and in the Introduction describes sex differences in alcohol-related responses - this seems like the necessary frame of reference from which to examine differences among cell means that were not large enough to drive a statistical interaction: planned comparisons.  The results should be presented as such. 

We have eliminated Treatment comparisons within each sex (unless there is a significant interaction). In the event of a significant main effect of Treatment (but not Sex) in the absence of an interaction, we have collapsed across Sex and performed Tukey’s post hocs. In the case of 2 significant main effects, we have not done further post hocs. We have also revised the title, abstract, results and discussion accordingly.

Reviewer 2 Report

Cortez et al., have significantly improved the manuscript taking into the previously suggested points. 
The data does not support the claims made in the paper.

Author Response

REVIEWER 2

Cortez et al., have significantly improved the manuscript taking into the previously suggested points. 
The data does not support the claims made in the paper.

We have modified the post hoc analyses and revised the title, abstract, results and discussion accordingly.